# Development of Experimental Techniques for the Phase Equilibrium Study in the Pb-Fe-O-S-Si System Involving Gas, Slag, Matte, Lead Metal and Tridymite Phases

**Taufiq Hidayat** [1,2] 🆔**, Ata Fallah-Mehrjardi** [1,3]**, Maksym Shevchenko** [1,]*🆔**, Peter C. Hayes** [1] **and Evgueni Jak** [1]

1 Pyrometallurgy Innovation Centre (PYROSEARCH), School of Chemical Engineering,
The University of Queensland, Brisbane, QLD 4072, Australia
2 Metallurgical Engineering Department, Faculty of Mining and Petroleum Engineering,
Institut Teknologi Bandung, Bandung 40132, Indonesia
3 Global Research, Vesuvius GH Rue de Douvrain 17, 7011 Ghlin, Belgium
* Correspondence: m.shevchenko@uq.edu.au

**Abstract:** Present society challenges, including metal scarcity, recycling, and environmental restrictions, resulted in the increased complexity and variability of metallurgical feed streams. Metallurgical processes involving complex lead and copper-containing slag and matte phases are now commonly used in response. Optimization of existing and development of new metallurgical processes require fundamental information on slag–matte phase equilibrium. Development of the experimental methodology for the characterization of slag–matte phase equilibrium is presented in the paper. Following a detailed analysis of the potential reaction pathways, experimental techniques have been developed that enable accurate measurement of slag–matte phase equilibrium in the Pb-Fe-O-S-Si system. The application of the techniques has been demonstrated for two important sets of conditions: (i) Condensed phase equilibrium for the slag–matte–metal–tridymite subsystem; and (ii) Gas–slag–matte–tridymite equilibrium at fixed oxygen and sulfur partial pressures. The experimental methodology involves high-temperature equilibration of synthetic samples, fast quenching, and microanalysis of the compositions of phases using electron probe X-ray microanalysis (EPMA). The experimental results are not affected by the changes in the bulk composition of samples during equilibration; this helps to overcome the significant barriers previously encountered in undertaking accurate measurement and characterization of these systems.

**Keywords:** lead slag; lead matte; lead metal; phase equilibria experiments

## 1. Introduction

As a result of the increasing complexity of ores, the incorporation of recycled materials of variable origin, the increasing concentrations of impurities and variability of the process streams [1], accurate fundamental information on the slag–matte equilibrium in lead-containing systems and the development of predictive tools are required for the optimization of the lead smelting, reduction, and recycling processes. A comprehensive integrated experimental and thermodynamic modeling overall research program focused on the characterization of the multi-component multi-phase gas–slag–matte–speiss–metal–solids system with the Pb-Zn-Cu-Fe-Si-Ca-O-S major, Al-Mg-Cr slagging, and As, Sn, Sb, Bi, Ag, Au, Ni, Co other minor elements [2,3] is carried out to address this demand. The outcomes of this overall research program include fundamental experimental measurements and the self-consistent thermodynamic database of the above multi-component multi-phase system. During this integrated research, predictions are used to assist in the planning of the experiments, and experimental results are used to continuously improve this internal database [4,5]. FactSage computer package [6,7] is used for thermodynamic modeling.

Fundamental experimental information on the phase equilibrium in the Pb-Fe-O-S-Si system—the focus of the study presented in this paper and undertaken as part of the overall research program [2,3]—is essential for understanding and improving the high-temperature metallurgical processing of lead and many recycling processes. Previous experimental studies of the lead-containing systems have provided data on the equilibria between the matte–metal [8–12], gas–slag [13–27], or slag–metal [21–30] phases. Only very limited information was found in the literature on the slag–matte equilibrium [28,29]. The lack of information on this system is due to the experimental difficulties of the accurate measurement of the equilibrium between slag and matte phases in this high-lead system associated with the high vapor pressures of the lead gaseous species, in particular PbS. Figure 1 shows the partial pressures of the Pb gaseous species as a function of the Pb concentration in slag at 1200 °C and 1 atm of total pressure (a) for the gas–slag–matte–tridymite equilibrium at fixed $p(SO_2) = 0.6$ atm and (b) for the slag–matte–metal equilibrium conditions corresponding to the lower $p(SO_2)$ from ~$10^{-6}$ to ~0.3 atm for the system with Pb in slag from ~2 to 50 wt.%Pb, predicted using the internal database [4,5] developed as a result of the present overall research program [2,3]. It can be seen that the partial pressures of the Pb-containing gaseous species are high, with the partial pressure $p(PbS)$ having the highest value followed by $p(Pb)$ and $p(PbO)$, and continuous lead loss from the sample to the atmosphere, therefore, will take place during the equilibration unless measures are taken to control or limit the Pb volatilization.

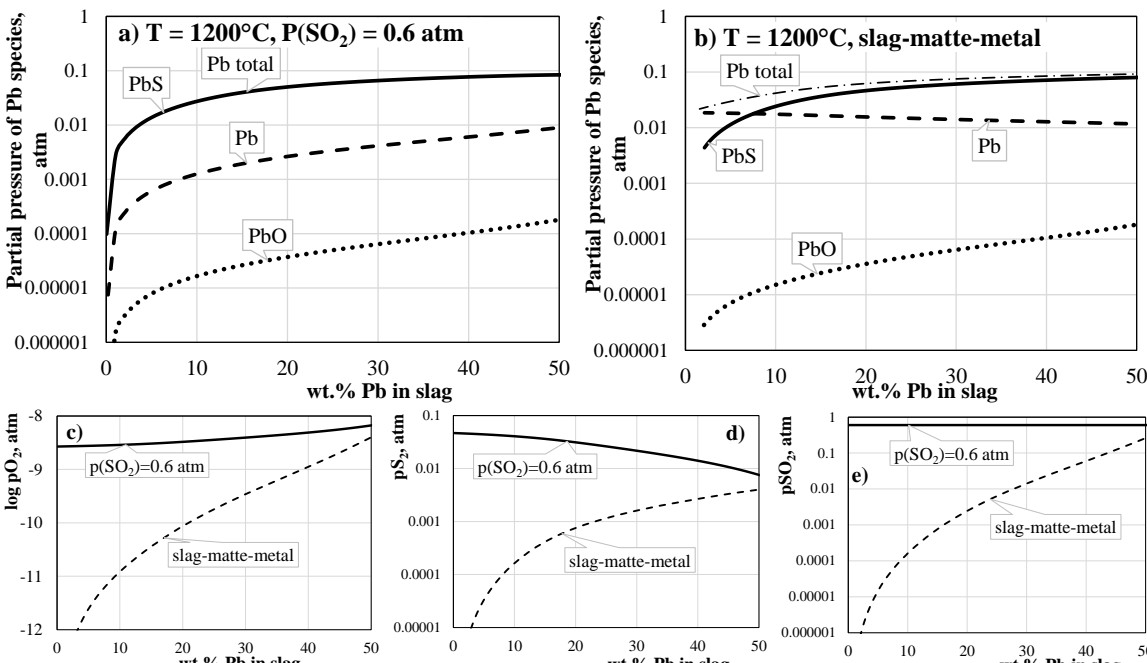

**Figure 1.** Calculated equilibrium partial pressures of Pb gas species at 1200 °C in the Pb-Fe-O-S-Si system, (**a**) $p(SO_2) = 0.6$ atm, slag–matte–tridymite, (**b**) at lower $p(SO_2)$, slag–matte–metal–tridymite, and associated (**c**) pO$_2$, atm, (**d**) pS$_2$, atm, (**e**) pSO$_2$, atm, using the FactSage internal thermodynamic database [4,5]. Note: in (**a**) Pb (total) overlaps with PbS.

The aim of the present paper is a detailed outline of the development of reliable experimental methodology for the accurate and systematic characterization of the gas–slag–matte–metal–solid equilibrium in the Pb-Fe-O-S-Si system at controlled conditions. The approach to the development of experimental methodology outlined in the present paper is generic and applicable to the studies of other similar complex systems. The methodology outlined in the present paper has been successfully applied for the investigation of the lead-containing gas–slag–matte–metal systems, including Pb-Fe-O-S-Si [30], Pb-Fe-O-S-Si-Cu [31], Pb-Fe-O-S-Si-(Al, Mg, Zn) [32].

Two critical conditions have been included in the present study: (i) slag–matte–lead metal–tridymite equilibrium (closed system in sealed ampoules, conditions controlled by the condensed phase equilibria), and (ii) slag–matte–metal–tridymite equilibrium at fixed $p(SO_2)$ = 0.6 atm (open system, conditions controlled by the gas phase). The two selected conditions are directly relevant to the commercial lead smelting, reduction, and recycling processes.

## 2. Materials and Methods

The general experimental approach used in the present study is based on the methodology developed by the co-authors [14,33] that involves high-temperature equilibration at controlled conditions, rapid quenching, and direct measurement of the compositions of equilibrium phases using electron probe X-ray microanalysis (EPMA). The synthetic samples were prepared from pure powders or pre-melted master slags or mattes, mixed in predetermined proportions, and pelletized. The initial compositions of the mixtures were selected so that one or more crystalline phase(s) would be present in equilibrium with liquid slag, matte, or metal. An iterative procedure involving preliminary experiments was generally needed to achieve the targeted proportions of the phases for a given final temperature and composition. Predictions with the preliminary internal thermodynamic database [4,5] were used to plan experiments.

Master slag and master matte with lead metal were used in the present study to facilitate faster equilibration. The master slag was prepared by mixing the Fe (99.9%) + $Fe_3O_4$ (99.9%) + $SiO_2$ (99.9%) pure powders (supplied by Alfa Aesar, Haverhill, MA, USA) and melting at 1350 °C under an inert argon atmosphere. Master matte–lead metal mixture was prepared by mixing pure Pb (99.99%), FeS (99.5%), and S (99.99%) powders (supplied by Alfa Aesar, Haverhill, MA, USA), melting the mixture in the enclosed graphite crucible with the graphite lid in a muffle furnace at 1200 °C for 30 min, and quenching in water. The average compositions of the master slag and master matte determined by measurements using EPMA are provided in Table 1.

**Table 1.** Average compositions (wt.%) of the master slag and master matte measured by EPMA.

| Sample | FeO | SiO$_2$ | |
|---|---|---|---|
| Master slag | 63.4 | 36.3 | |
| Sample | Pb | Fe | S |
| Master matte | 62.6 | 17.9 | 19.5 |
| Master metal | 98.4 | 0.1 | 1.5 |

The initial mixtures for the experiments were prepared from the master slag and the master matte + lead metal mixtures with the addition of pure powders, then pelletized and placed in silica ampoules. Two types of experiments were undertaken:

(i)     Closed experiments—in sealed silica ampoules under an argon atmosphere;
(ii)    Semi-open experiments—in silica ampoules in a fixed $CO/CO_2/SO_2/Ar$ gas mixture (all gases 99.999% purity, supplied by Coregas, Yennora, Australia).

The ampoules containing the samples were then suspended on a Kanthal (Fe-Cr-Al) wire and equilibrated in a vertical impervious alumina ceramic tube in an electrically-heated resistance furnace. A calibrated Pt/Pt-Rh 13% thermocouple protected by the alumina shield was placed immediately adjacent to the sample to directly monitor the equilibration temperature. The working thermocouple was calibrated against a standard thermocouple (supplied by the National Measurement Institute of Australia, NSW, Australia). The overall absolute temperature accuracy of the experiment was estimated to be ±3 K. After equilibration for selected times, the samples were quenched into cold water and mounted in epoxy resin. A copper plate was placed under a ~10 mm layer of cold water to facilitate

breakage of the ampoules falling from the furnace on quenching to achieve fast cooling rates (see Figure 2).

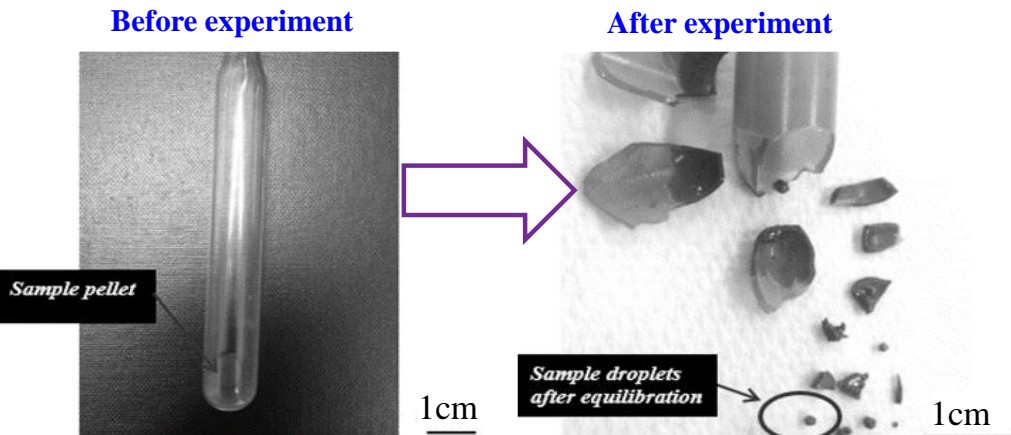

**Figure 2.** Closed-system equilibration used to study the slag–matte–lead–tridymite equilibrium in the Pb-Fe-O-S-Si system.

Polished cross-sections of the samples were prepared using standard metallographic techniques. The compositions of the individual phases in the samples were measured using EPMA JEOL JXA 8200L (trademark of Japan Electron Optics Ltd., Tokyo, Japan) at an acceleration voltage of 15 kV and a probe current of 15 nA. Oxide ($SiO_2$, $Fe_2O_3$), sulfide (PbS, $FeS_2$), and metal (Fe) standards from Charles M. Taylor, Stanford, CA, USA, were used. The standard Duncumb–Philibert ZAF correction supplied by JEOL was applied. Iron is always present in the slag in the $Fe^{2+}$ and $Fe^{3+}$ forms. Only the concentrations of the metal cations in the phases were measured with EPMA in the present study without information on their oxidation states. All concentrations of the metal cations in the liquid and solid oxides from the EPMA analysis were recalculated to selected oxidation states (i.e., "PbO", "FeO", and $SiO_2$) for presentation purposes only to unambiguously report the compositions of the phases. A non-zero probe diameter of up to 100 μm was used to measure slag, matte, and metal compositions due to the formation of microcrystalline precipitates in these phases during quenching.

A four-points-test approach [33] was used in the present study to identify possible uncertainties and to develop suitable techniques for the particular investigated system to ensure the equilibrium is achieved. The four-points-test includes:

(i)   Analysis of the effect of equilibration time,
(ii)  Examination of the homogeneity of the equilibrium phases,
(iii) Evaluation of the effect of the direction of approach toward the equilibrium point, and
(iv)  Investigation of the behavior of the system through systematic analysis and identification of the elementary reactions occurring during the equilibration and their effects on the achievement of equilibrium, which are specific to the system under investigation.

### 3. Systematic Analysis of the Elementary Reactions

*3.1. Systematic Analysis of the Reactions Taking Place during Equilibration in the Pb-Fe-O-S-Si System*

An equilibration process in the multi-component multi-phase system involves different elemental reactions that can affect, inhibit or completely block the achievement of equilibrium. These elemental reactions directly depend on (1) the phases present in the initial mixture, (2) starting composition, (3) the proportion of phases, (4) the heat treatment regime, (5) sample size, (6) containment, and, therefore, can be controlled by the adjustment of these experimental conditions. Systematic analysis of all important elementary reactions between chemical components within a single phase and between different phases, therefore, is essential to identify those possible elementary reactions that limit the achievement of equilibria and, consequently, to modify the experimental methodology (including the

parameters 1–6 above). This is particularly important for the present system in which equilibrium between several phases must be achieved, and one of the components has high vapor pressure. Based on this preliminary systematic hypothetical analysis of elementary reactions, microanalysis of compositional trends in the preliminary experiments can be used to identify the actual reactions taking place during equilibration, the corresponding developments of specific methodology suitable to the particular system can be introduced and implemented, and the achievement of equilibrium can be tested and confirmed. This preliminary analysis is outlined in the following sections.

The phases, the predominant chemical species present in each phase, and possible key chemical interactions between the gas, slag, matte, metal, and tridymite phases in the Pb-Fe-O-S-Si system taking place during equilibration are illustrated in Figure 3 and summarised in Table 2. The elementary reaction steps can take place within (I) a single phase or between (II) two, (III) three, and (IV) four phases.

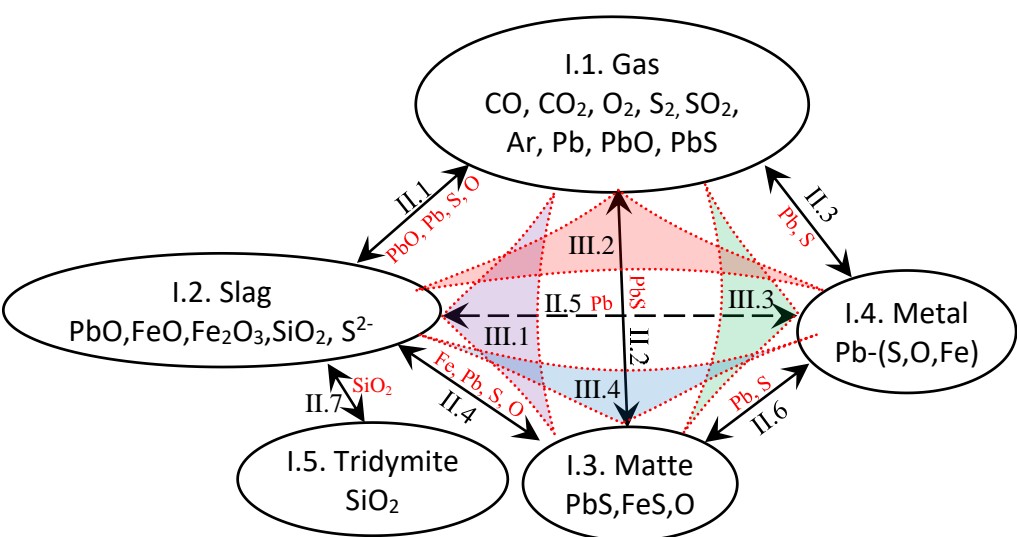

**Figure 3.** Schematic of possible interactions of components between the gas, slag, matte, metal, and tridymite phases in the Pb-Fe-O-S-Si system, including: I. Single, II. Two-, III. Three-, and IV. Four-phase interactions. Details of the reactions are given in Table 2.

**Table 2.** Summary of selected elementary reactions and reaction steps taking place in the Pb-Fe-O-S-Si system for I. Single, II, Two-, III, Three-, and IV Four-phase interactions (indications of the most important reactions for the closed and open systems are given in the marked columns). Note: g = gas; sl = slag; mat = matte; met = metal.

| Locations | Reactions | Closed | Open |
|---|---|---|---|
| I.1. Gas | (a) Mass transfer of gaseous species to/from the sample surface (in gas bulk or through film layer) | | |
| | (b) Reaction between gas species: | | |
| | i. $CO_2$ (g) = $1/2$ $O_2$ (g) + CO (g) | | + |
| | ii. $SO_2$ (g) = $O_2$ (g)+ $1/2$ $S_2$ (g) | | |
| | iii. $SO_2$ (g) + 2CO (g) = $1/2$ $S_2$ (g) + $2CO_2$ (g) | | |
| I.2. Slag | (a) Diffusion of Pb, $Fe^{2+}$, $Fe^{3+}$, Si, S and O within liquid slag | + | + |
| | (b) Oxygen transfer through slag taking place by the ferric–ferrous couple: 2FeO (sl) + [O] (sl) = $2FeO_{1.5}$ (sl) | | |
| I.3. Matte | Diffusion of Pb, Fe, O, and S within matte | | |
| I.4. Metal | Diffusion of Pb, (Fe), O, and S within metal (Fe is very low) | | |

**Table 2.** *Cont.*

| Locations | Reactions | Closed | Open |
|---|---|---|---|
| II.1. Gas–slag | (a) Adsorption/chemical reaction/desorption at gas–slag interface | | |
| | (b) Oxidation/reduction of slag $2FeO_{1.5}$ (sl) + CO (g) = 2FeO (sl) + $CO_2$ (g) | | + |
| | (c) PbO (sl) + CO (g) = Pb (g) + $CO_2$ (g) (can be important since $P_{Pb} >> P_{PbO}$—see Figure 1) | | + |
| | Sulfurization/de-sulfurization of slag: | | |
| | (d) $SO_2$ (g) + 3CO (g) + FeO (sl) = FeS (sl) + $3CO_2$ (g) | | |
| | (e) 7FeO (sl) + $SO_2$ (g) = $6FeO_{1.5}$ (sl) + FeS (sl) | | |
| | Evaporation of slag component: | | |
| | (f) PbO (sl) = PbO (g) | | |
| | (g) 2FeO (sl) + PbO (sl) = Pb (g) + $2FeO_{1.5}$ (sl) | | |
| II.2. Gas–matte | (a) Adsorption/chemical reaction/desorption at gas–matte interface | | |
| | (b) Oxygen and sulfur adsorption/dissolution in matte: | | |
| | i. $1/2\ O_2$ (g) = [O] (mat) | | |
| | $CO_2$ (g) = [O] (mat) + CO (g) | | |
| | ii. $SO_2$ (g) = S (mat) + $O_2$ (g) | | + |
| | $SO_2$ (g) + CO (g) = S (mat) + $CO_2$ (g) | | + |
| | $1/2\ S_2$ (g) = S (mat) | | |
| | (c) Evaporation of matte component: PbS (mat) = PbS (g) | | + |
| II.3. Gas–metal | (a) Adsorption/chemical reaction/desorption at gas–metal interface | | |
| | (b) Oxygen and sulfur adsorption/dissolution in metal: | | |
| | i. $1/2\ O_2$ (g) = [O] (met) | | |
| | $CO_2$ (g) = [O] (met) + CO (g) | | + |
| | ii. $SO_2$ (g) = S (met) + $O_2$ (g) | | |
| | $SO_2$ (g) + CO (g) = S (met) + $CO_2$ (g) | | + |
| | iii. Pb (met) + $1/2\ O_2$ (g) = PbO (g) | | |
| | Pb (met) + $CO_2$ (g) = PbO(g) + CO (g) | | + |
| | iv. Pb (met) + $SO_2$ (g) + 2CO (g) = PbS (g) + $2CO_2$ (g) | | + |
| | (c) Evaporation of metal component: Pb (met) = Pb (g) | | + |
| | (d) Pb (met) + S (met) = PbS (g) | | |
| II.4. Slag–matte | (a) FeS (mat) + PbO (sl) = PbS (mat) + FeO (sl) | + | + |
| | (b) FeS (mat) = FeS (sl); PbS (mat) = PbS (sl) | | |
| | (c) 2FeO (sl) + [O] (mat) = $2FeO_{1.5}$ (sl) | | |
| II.5. Slag–metal | (a) PbO (sl)+ 2FeO (sl) = $2FeO_{1.5}$ (sl) + Pb (met) | + | + |
| | (b) $2FeO_{1.5}$ (sl) + [Fe] (met) = 3FeO (sl) | | |
| | (c) 2FeO (sl) + [O] (met) = $2FeO_{1.5}$ (sl) | | |
| II.6. Matte–metal | (a) Matte–metal exchange reactions PbS (mat) = Pb (met) + [S] (met) | + | + |
| | (b) FeS (mat) + Pb (met) = PbS (mat) + Fe (met) | | |
| | (c) cross-boundary diffusion S (mat) = [S] (met); [O] (mat) = [O] (met); Pb (mat) = Pb (met) | + | + |
| II.7. Slag–Tridymite | Dissolution/precipitation of tridymite into/from slag: $SiO_2$ (solid) = $SiO_2$ (sl) | + | + |
| III.1. Gas–slag–matte | (a) PbS (mat) + 3 $CO_2$ (g) = PbO (sl) + $SO_2$ (g) + 3CO (g) | | + |
| | (b) FeS (mat) + 3 $CO_2$ (g) = FeO (sl) + $SO_2$ (g) + 3CO (g) | | + |
| | Combined ab: $\alpha$PbS (mat) + $\beta$FeS (mat) + 3($\alpha$ + $\beta$)$CO_2$ (g) = $\alpha$PbO (sl) + $\beta$FeO (sl) + 3($\alpha$ + $\beta$)CO (g) + ($\alpha$ + $\beta$)$SO_2$ (g) | | + |
| | (c) PbS (mat) + $6FeO_{1.5}$ (sl) = PbO (sl) + 6FeO (sl) + $SO_2$ (g) | | + |
| | (d) FeS (mat) + $6FeO_{1.5}$ (sl) = 7FeO (sl) + $SO_2$ (g) | | + |
| | (e) FeS (mat) + $FeO_{1.5}$ (sl) = 2FeO (sl) + $1/2\ S_2$ (g) | | + |

**Table 2.** *Cont.*

| Locations | Reactions | Closed | Open |
|---|---|---|---|
| III.2. Gas–slag–metal | (a) PbO (sl) + CO (g) = Pb (met) + $CO_2$ (g) | | + |
| | (b) FeO (sl) + CO (g) = Fe (met) + $CO_2$ (g) (Fe is very low) | | |
| III.3. Gas–matte-Metal | (a) Pb (met) + 2CO (g) + $SO_2$ (g) = PbS (mat) + $2CO_2$ (g) | | |
| | (b) Pb (met) + FeS(mat) = PbS (g) + Fe(met) (Fe is very low) | | |
| III.4. Slag–matte–metal | (a) PbS (mat) + $2FeO_{1.5}$ (sl) = PbO (sl) + 2FeO (sl) + [S] (met) | | |
| | (b) FeS (mat) + $2FeO_{1.5}$ (sl) = 3FeO (sl) + [S] (met) | | |
| | (c) FeS (mat) + $2FeO_{1.5}$ (sl) + Pb (met) = PbS (mat) + 3FeO (sl) | + | + |
| | (d) PbS (mat) + $2FeO_{1.5}$ (sl) + Pb (met) = FeS (mat) + FeO (sl) + 2PbO (sl) | + | + |
| IV.1. Gas–slag–matte–metal | 4-phase reactions (e.g., 3Pb (met) + $SO_2$ (g) = PbS (mat) + 2PbO (sl)) are not important for the present study of the slag–matte–metal–tridymite and slag–matte–metal–tridymite systems | | + |
| | No direct reaction or mass exchange through IV.2. Gas–slag–matte–tridymite; IV.3. Gas–slag–metal–tridymite; IV.4. Gas–matte–metal–tridymite; or IV.5. Slag–matte–metal–tridymite is expected | | |

The general principles used to identify key reactions possibly taking place in the system are based on the phases present, compositional limits of the elements in each phase, and predominant species in each phase that determine (a) what components can react within each phase and between phases, and (b) what components can be transferred by diffusion and convective mass transfer through each phase. The approach is (1) to include all relevant reactions and the (2) to limit/reduce their list.

Brief comments on each phase are given below.

Gas Phase: The predominant species present in the gas phase in the open system experiments are CO, $CO_2$, and $SO_2$, which are used to maintain selected $p(O_2)$ and $p(S_2)$; Ar is a neutral non-reactive species. Pb, PbO, and PbS species are present due to vaporization from the condensed phases. $SO_2$, $S_2$, and $O_2$ present in the closed system experiments at the partial pressures are determined by the condensed phases and their compositions.

Slag Phase: The liquid slag phase contains principally PbO, FeO, $FeO_{1.5}$, and $SiO_2$; S is present as a minor species. The 2FeO + $1/2\ O_2$ = $2FeO_{1.5}$ reaction either follows the $p(O_2)$ established by $CO/CO_2$ in the open system experiment or imposes the effective $p(O_2)$ in the closed system. Relatively low S concentrations in the slag results in relatively slow rates of mass transfer of this species through the slag phase.

Matte Phase: The compositions of the liquid matte phase in this system are close to the PbS-FeS join. Significant Pb and Fe mass transfer between the matte and other phases is expected to involve PbS and FeS reaction and/or diffusion and convection. Relatively low oxygen concentrations in the matte phase are expected to result in relatively low rates of mass transfer of this element within the matte and between the matte and other phases. The negligible concentration of Si in matte will result in complete blockage of Si mass transfer through the matte phase.

Lead Metal: The liquid lead metal phase contains relatively low concentrations of dissolved Fe, O, S, and negligible Si solubility. The rates of mass transfer of these elements through the metal phase are low relative to other pathways. Variation of the metal phase composition as a function of Pb in slag is not significant compared to those in the slag and matte phases. The metal phase acts as a source or "sink" of Pb for the slag and matte phases.

Tridymite: It is essentially pure stoichiometric $SiO_2$ (solubilities of Fe, Pb, and S are negligible). The tridymite particles are dispersed throughout the slag and also form a thick crystalline layer on the original amorphous $SiO_2$ substrate. The tridymite phase acts as a source or "sink" for $SiO_2$ in the slag.

These considerations are used to develop the list of reactions in Table 2, illustrated in Figure 3.

Further discussions are therefore focused only on the slag and matte phase compositional changes. Only the slag liquid solution has a significant concentration of $SiO_2$, so only the slag–tridymite reaction and mass transfer within the slag phase are considered for $SiO_2$. In summary, significant mass transfer of the following elements and species is expected to take place as a result of the chemical reactions: II.1. gas–slag—PbO, Pb, S, O; II.2. gas–matte—PbS; II.3. gas–metal—Pb, S; II.4. slag–matte—Fe, Pb, S, O; II.5. slag–metal—Pb; II.6. matte–metal—Pb, S; and II.7. slag–tridymite—$SiO_2$. These interactions and key reactions that are expected to be important are listed in Table 2 and summarised in Figure 3.

In the slag–matte–metal–tridymite (closed) system experiments carried out in sealed ampoules, the gas atmosphere is controlled by the condensed phases, and the overall mass transfer within and between condensed phases is dominant. In the slag–matte–metal–tridymite (open) system experiments carried out in semi-open ampoules, the gas atmosphere controls oxygen and sulfur dioxide partial pressures; however, the mass transfer within and between condensed phases is still important.

The following Sections 3.2 and 3.3 will provide a more detailed analysis and description of the reactions expected to occur in each particular case (a) for the closed and (b) for the open systems experiments. Three schematic diagrams are introduced for the analysis of each (a) closed and (b) open system.

1. The inter-phase mass transfer schematic diagram summarizes the key elements of mass transfer and key reactions (Figures 4 and 7). The major elements comprising each phase, the key elements that can transfer between the phases, the key reactions between phases that are expected to be important, and the limiting factors that may block those reactions are identified for all combinations of the II- and III- phase assemblages (gas, slag, matte, metal), and separately—for the II-7 slag—tridymite phase assemblage.

2. The compositional change direction diagram shows the direction of compositional changes in key phases for each reaction (Figures 5 and 8). For each type of experiment (closed and open), possible changes of the matte and slag compositions are plotted on the $Pb_{matte}$ vs. $Pb_{slag}$ and PbO-"FeO"-$SiO_2$ diagrams, and the vectors of the compositional changes corresponding to each reaction are indicated on those diagrams.

3. The micro/macro location diagram identifies the typical locations where the reactions take place within the samples (Figures 6 and 9). The classification of the different typical micro- and macro-locations of the gas, slag, matte, metal, and solid phases in the sample relative to each other and to the position in the sample (e.g., relative to the gas-condensed phase interface or the sample-substrate interface) are summarised to facilitate the systematic analysis of the reactions using the EPMA measurements of compositional profiles and trends. The classification is based on the following criteria:

    (i) The locations of the slag, matte, and metal phases relative to each other and relative to their positions within the sample (close to the substrate or to the gas-condensed phase interface),
    (ii) The morphology (shape) of the matte and metal particles, and
    (iii) Size of the matte and metal phases.

The (1) interphase mass transfer, (2) compositional change direction diagrams, and (3) the classification of locations are essential to:

(i) Systematically analyzing the compositional changes/profiles at the micro- and macro-scales using EPMA;
(ii) Identifying the possible reactions taking place between phases and within phases;
(iii) Linking the expected and actual measured experimental points relative to equilibrium values in different phases to identify and confirm the reactions taking place during equilibration, and, therefore,
(iv) Introducing modification and improvements to the experimental methodology; and finally,
(v) Confirming the achievement of equilibrium.

The effects of the location on some reactions are considered; for example, large phases blocking mass transfer between small phases, or large phases dominating the conditions of the small phases encapsulated inside them.

An example of the development and application of such classification is given by Fallah-Mehrjardi et al. [34] for the Cu-containing gas–slag–matte–metal–solid system.

The detection of incomplete reactions in the experimental study during equilibration, starting from the initial condensed phase compositions and progressing toward the equilibrium phase compositions by measurement of the macro- and micro-inhomogeneities in the phases in the quenched samples using microanalysis techniques, such as EPMA, which has less than 1 μm take-off volume in this chemical system, provides essential information for the development of a methodology that ensures the equilibrium is achieved. The macro- and micro-inhomogeneities are defined as follows:

- The macro-inhomogeneities are usually identified on the scale from ~50 μm to several millimeters across the sample relative to the gas-condensed interface, between large phases (>100 μ) or relative to the substrate interfaces, and
- The micro-inhomogeneities are usually identified on the scale ~5–50 μm across one phase (e.g., slag) relative to the distance from the phase boundaries.

It is this ability to identify the reactions by the measurement of the macro- and micro-compositional trends using microanalysis techniques, such as EPMA, that is the strength of the present experimental methodology.

An important assumption used in such analysis is that the initial chemicals react relatively rapidly within a few minutes at the start of the experiment forming condensed phases (compared to the relatively slower reactions between the gas and condensed phases) so that it is possible to approximately predict the initial composition of each condensed phase at the start of the experiment. The initial phase compositions are then confirmed in the preliminary of very short (≤15 min) experiments by the EPMA measurements of samples.

### 3.2. Reactions Taking Place during Closed System, Slag-Matte–Lead Metal-Tridymite Equilibration

Figure 4 illustrates schematically the key reactions expected to take place in the closed system, including:

Slag–matte—II.4.a: FeS (mat) + PbO (sl) $\leftrightarrow$ PbS (mat) + FeO (sl),

Slag–metal—II.5.a: PbO (sl) + 2FeO (sl) $\leftrightarrow$ 2FeO$_{1.5}$ (sl) + Pb (met), limited by Fe$^{2+}$/Fe$^{3+}$ in slag,

Matte–metal—II.6.a: PbS (mat) $\leftrightarrow$ Pb (met) + [S] (met), limited by S in metal,

Slag–tridymite—II.7: SiO$_2$ (solid) $\leftrightarrow$ SiO$_2$ (sl), adjusting toward the tridymite liquidus,

Slag–matte–metal—III.4.c: FeS (mat) + 2FeO$_{1.5}$ (sl) + Pb (met) $\leftrightarrow$ PbS (mat) + 3FeO (sl), and

III.4.d: PbS (mat) + 2FeO$_{1.5}$ (sl) + Pb (met) $\leftrightarrow$ FeS (mat) + FeO (sl) + 2PbO (sl).

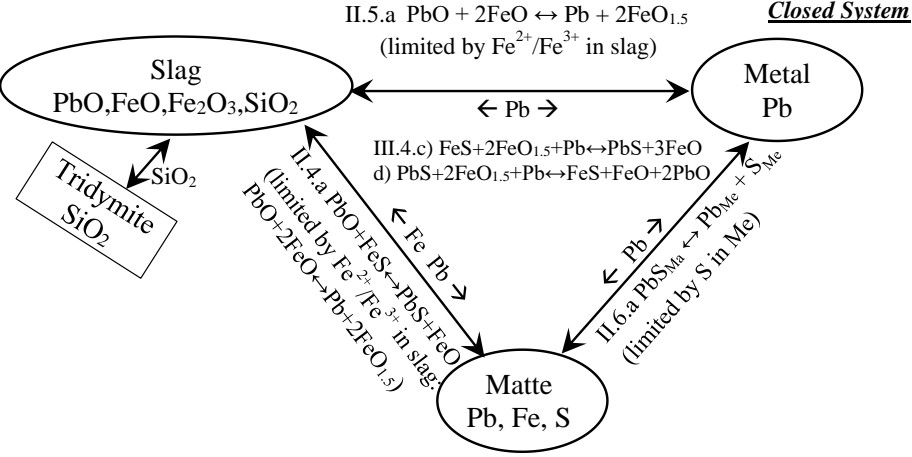

**Figure 4.** Schematic inter-phase mass transfer diagram of the key mass transfer of elements between phases and reactions, expected to be important in the closed system.

The vectors corresponding to the compositional changes resulting from these reactions are plotted in Figure 5 on the $Pb_{matte}$ vs. $Pb_{slag}$ graph to illustrate the slag–matte relative compositional changes and on the PbO-"FeO"-SiO$_2$ phase diagram to illustrate the slag compositional changes relative to the tridymite liquidus. As indicated above, the metal phase compositional changes are not included in this graphical analysis.

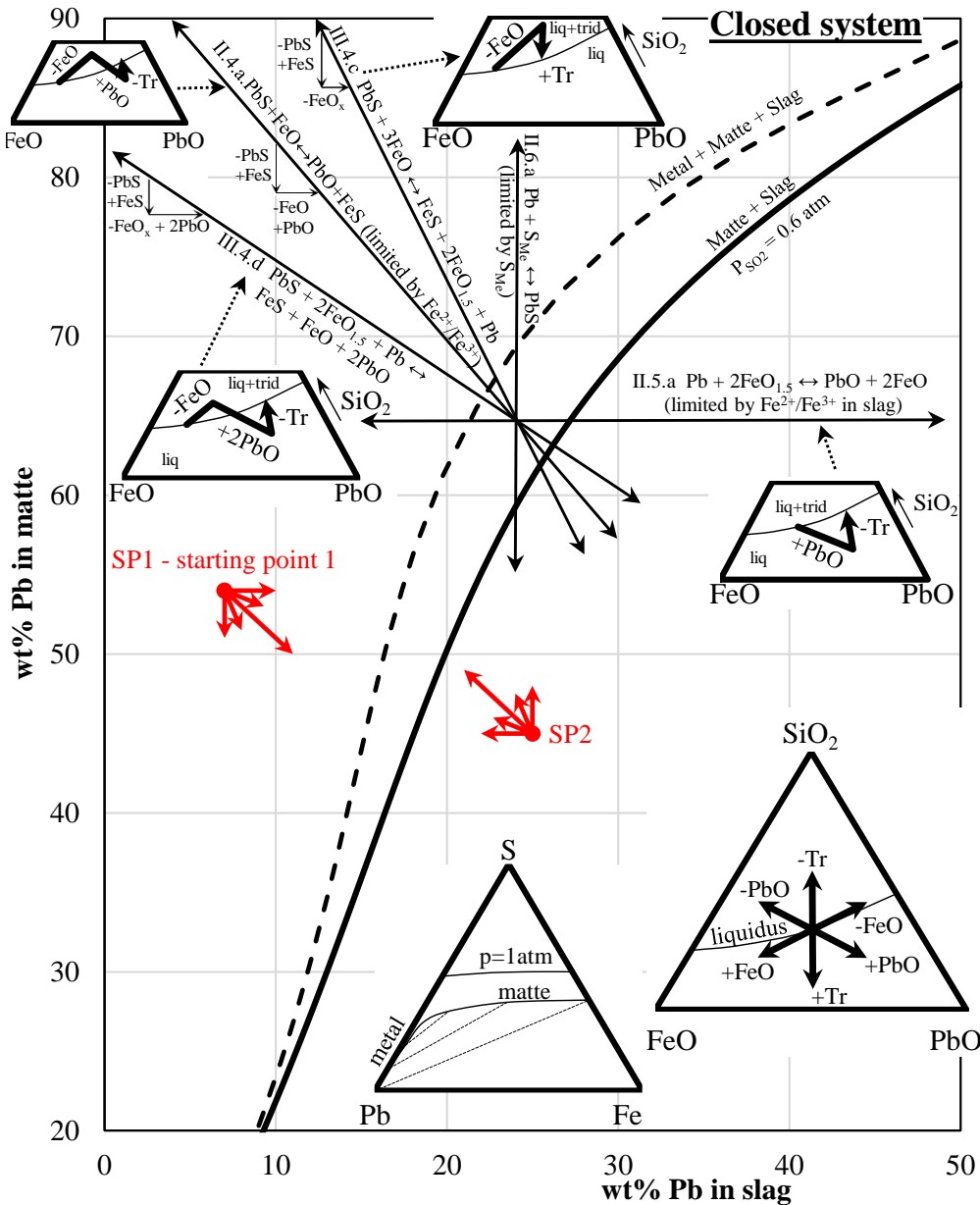

**Figure 5.** The compositional change direction diagrams for $Pb_{matte}$ vs. $Pb_{slag}$ and PbO-"FeO"-SiO$_2$ with the vectors showing the directions of the compositional changes in matte and slag phases for reactions in the closed system.

As a result of the reaction, stoichiometry reduction by one mole of FeS from the matte phase is equivalent to the addition of one mole of PbS and is indicated using the notation "$-$FeS" = "+PbS". The slag composition changes due to the reduction or addition of one mole of FeO, FeO$_{1.5}$, or PbO are indicated using the notation $\pm$FeO (indicating projection of both $\pm$FeO and $\pm$FeO$_{1.5}$ onto the "FeO"-PbO-SiO$_2$ plane) or $\pm$PbO, respectively.

The composition changes of slag, in addition to the reactions with matte and metal, are also directed by the slag–tridymite phase equilibrium. If the FeO, FeO$_{1.5}$, and PbO are transferred into the slag and, thus, change its composition away from the SiO$_2$ apex toward

the PbO or "FeO" apexes, dissolution of tridymite into the slag will take place, moving the slag composition back toward the tridymite liquidus. If the FeO, $FeO_{1.5}$, and PbO are taken from the slag, the tridymite precipitation will restore phase equilibrium, moving the slag composition back toward the liquidus isotherm. Since the precipitation of tridymite (Tr) from the liquid slag is opposite to the dissolution of tridymite into the liquid slag, these reactions are indicated by the notation $-$Tr $=$ $+$Tr, respectively. The corresponding changes of the slag composition projected onto the PbO-"FeO"-$SiO_2$ diagram are also included in Figure 5.

The $Pb_{matte}$ vs. $Pb_{slag}$ trend corresponding to the closed slag–matte–metal–tridymite equilibrium at 1200 °C and 1 atm of total pressure calculated with the current preliminary FactSage internal database [4,5] is plotted in Figure 5 with the dashed line. The initial compositions starting points SP1 and SP2 indicated in Figure 5 are arbitrarily selected in the areas above and left and below, and right of the equilibrium conditions, respectively, for the presentation purposes to hypothetically explain and to analyze possible changes of compositions during equilibration in order to assist in further identification of important reactions. The initial compositions of the slag and matte phases in samples, such as SP1 at higher Pb in matte and lower Pb in slag, relative to the equilibrium compositions, would move to the right and down on the $Pb_{matte}$ vs. $Pb_{slag}$ diagram as a result of the reactions (see Figure 5). The phases in the initial sample SP2 would move to the left and up on the $Pb_{matte}$ vs. $Pb_{slag}$ diagram toward the equilibrium trend indicated in the figure by the dashed line for the case of the closed slag–matte–metal–tridymite system.

The schematic given in Figure 6 illustrates the classification of macro- and micro-locations in the closed system—the important typical combinations of phase sizes and locations derived from the above analysis of the expected reactions and mass transfer processes for the closed system experiments. Large–large, large–small, and small–large combinations of phase sizes are considered for the reactions between two particular phases indicated with the reaction number "II.x." and symbols "l-l", "l-s," and "s-l", following the 1. slag—2. matte—3. metal sequence of phases, respectively (note that the small–small combination is excluded from analysis as not relevant in this study). For example, the large slag—small matte combination of phases is indicated as "II.4.l-s" (where "II.4" indicates the slag–matte reactions—see Table 2). Similarly, eight types of locations were identified for the three-phase slag–matte–metal reactions, indicated by "III.4" (see Table 2)—l-l-l, l-s-l, l-l-s, l-s-s, s-l-l, s-s-l, s-l-s, and s-s-s. Each of the slag-containing phase combinations can have tridymite present or not present in the vicinity of the phase location indicated by "(t)" or "(n)", respectively. Tridymite in the vicinity of a particular phase combination location is important to provide $SiO_2$ into slag in case of dissolution trend and to act as a nucleation site in case of $SiO_2$ precipitation trend.

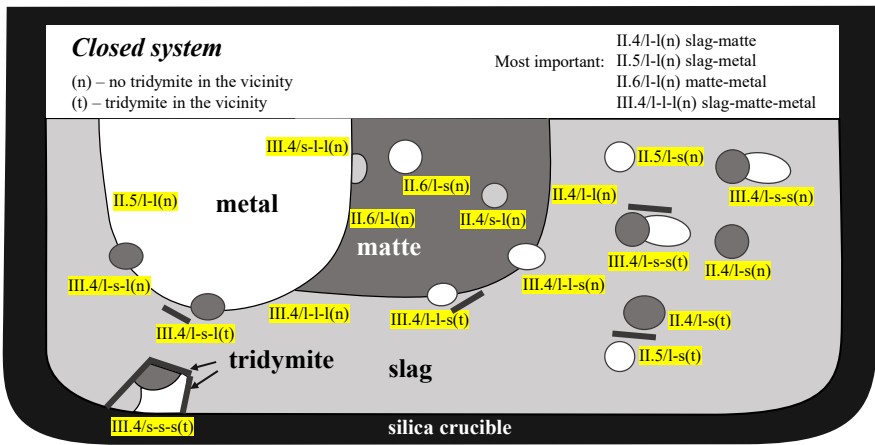

**Figure 6.** Schematic diagram for classification of macro- and micro-locations in the closed system.

This classification system enables the outcomes of phase interactions during equilibration to be systematically analyzed, and hypotheses developed. The following statements will refer to relatively short-time preliminary experiments in which the overall equilibrium across the whole sample at the macro-scale was not achieved, with only local equilibria achieved in different parts of samples at the micro-scale. Identification of such non-equilibrium locations at the preliminary stage confirms the key reactions and, therefore, allows to make necessary adjustments to the experimental methodology.

The following considerations are important to analyze possible reaction trends and to develop hypotheses.

Since the metal is expected to have low mass transfer rates of S, Fe, and O, it will act (i) as the source or "sink" of Pb for those reactions in which the phases are in direct contact with metal and (ii) as a barrier phase, blocking the reactions between other phases that are separated by a large mass of the metal phase (see Figure 6).

Small particles of slag are observed to be very rarely encapsulated in the large metal or matte particles as a result of a combination of physical properties, such as density, surface tension, etc.

Large matte particles may contain small suspended slag or metal droplets (e.g., locations II.6/l-s(n) and II.4/s-l(n)), which are expected to be blocked from chemical interactions with other phases and can be far from the overall equilibrium and close to the initial compositions SP1 or SP2 compared to other locations. For example, the composition of small droplets of slag suspended in matte (II.4/s-l(n)) is dominated by matte, while small matte in slag (II.4/l-s(n)) is dominated by slag.

Small matte and metal droplets can commonly be located at the interfaces between large slag and metal phases and between large slag and matte droplets, respectively. The locations III.4/l-s-l(n) and III.4/l-l-s(n) can achieve the slag–matte–metal local equilibrium but not the equilibrium with tridymite. The locations III.4/l-s-l(t) and III.4/l-l-s(t) with tridymite in the vicinity can potentially achieve the slag–matte–metal–tridymite local equilibrium.

Large slag masses commonly contain suspended small matte and metal droplets, as well as tridymite crystals. The compositions of small matte or metal droplets usually cannot be accurately measured by EPMA if their size is smaller than several probe diameters (~100–200 microns). The locations II.5/l-s(n), II.4/l-s(n), II.5/l-s(t), II.4/l-s(t), and III.4/l-s-s(n) would be closer to the initial starting compositions SP1 or SP2, further away from the equilibrium condition. The conditions in the III.4.l-s-s(t) and III.4.l-l-l(t) locations having tridymite in the vicinity are expected to be closer to or at the equilibrium and to be different from the III.4.s-s-l(n) and III.4.l-l-l(n) locations without tridymite in the vicinity.

The differences in phase compositions between these locations measured with EPMA can indicate if a hypothesis on the particular reaction taking place is valid or not. The achievement of equilibrium in the III.4.l-l-l(t) location may still be affected by the PbS-FeS mass transfer in the large matte phase, so the conditions in the III.4.l-s-s(t), III.4.l-s-l(t), and III.4.s-s-s(t) locations are expected to be closer to the equilibrium. The definite non-equilibrium locations (such as III.4/s-s-l(n) and III.4/l-l-l(n) outlined above) in the preliminary experiments, being incompletely equilibrated, are used to confirm the hypotheses about the reactions.

### 3.3. Reactions Taking Place during Open System, Slag–Matte–Metal–Tridymite Equilibration

The following section presents a similar analysis for the open system experiments. The important difference between closed and open systems, in addition to the absence of the metal phase, is that in the open type of experiments, the evaporation of Pb and S from the condensed phases into the gaseous phase is an important factor.

Figure 7 schematically illustrates the vectors corresponding to the key reactions expected to take place in the open system, including:

Gas–slag—II.1.b Oxidation/reduction of slag $2FeO_{1.5}$ (sl) + CO (g) $\leftrightarrow$ 2FeO (sl) + $CO_2$ (g),
II.1.c PbO (sl) + CO (g) $\leftrightarrow$ Pb (g) + $CO_2$ (g),
II.1.g 2FeO (sl) + PbO (sl) $\leftrightarrow$ Pb (g) + $2FeO_{1.5}$ (sl), limited by $Fe^{2+}/Fe^{3+}$ in slag,

Gas–matte—II.2.c. Evaporation PbS (mat) ↔ PbS (g), unlimited, continuous,
Slag–matte—II.4.a FeS (mat) + PbO (sl) ↔ PbS (mat) + FeO (sl),
Slag–tridymite—II.7. SiO$_2$ (solid) ↔ SiO$_2$ (sl), adjusting toward tridymite liquidus,
Gas–slag–matte—III.1.a PbS (mat) + 3CO$_2$ (g) ↔ PbO (sl) + SO$_2$ (g) + 3CO (g),

III.1.b FeS (mat) + 3CO$_2$ (g) ↔ FeO (sl) + SO$_2$ (g) + 3CO (g),
III.1.c PbS (mat) + 6FeO$_{1.5}$ (sl) ↔ PbO (sl) + 6FeO (sl) + SO$_2$ (g), and
III.1.d FeS (mat) + 6FeO$_{1.5}$ (sl) ↔ 7FeO (sl) + SO$_2$ (g).

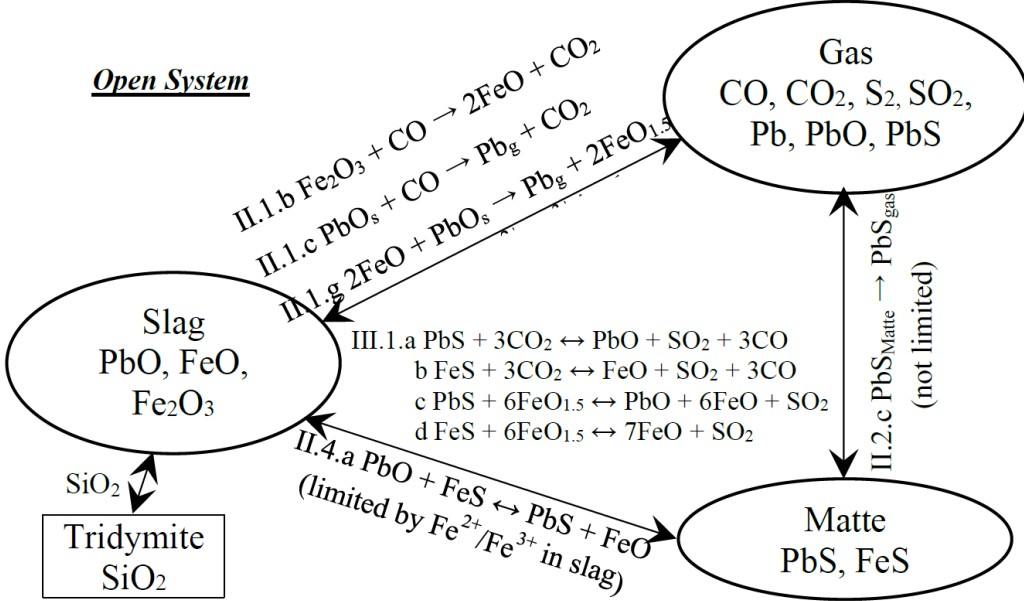

**Figure 7.** Interphase mass transfer diagram of the key mass transfer elements and reactions expected to be important in the open system.

The relative compositional changes in slag and matte resulting from these reactions are plotted in Figure 8 on the Pb$_{matte}$ vs. Pb$_{slag}$ diagram. The corresponding changes in the slag relative to the tridymite liquidus are plotted on the PbO-"FeO"-SiO$_2$ phase diagram. Initial compositions indicated with the starting points SP3 and SP4 in Figure 8 are arbitrarily selected above and left and below and right of the equilibrium conditions for the hypothetical analysis of possible compositional changes. Similar to the closed system described above, the overall directions of the compositional changes are shown with the notations ±FeS, ±PbS, ±FeO, ±FeO$_{1.5}$, ±PbO, and ±T. These reactions will result in compositional changes from the initial starting composition points SP3 and SP4 toward the equilibrium indicated in the figure by solid line predicted with the current preliminary FactSage internal database [4,5]. The initial composition of each condensed phase during the analysis of the reactions (e.g., indicated with SP3 and SP4 in Figure 8) is hypothetically predicted based on the assumption that the initial chemicals forming a particular condensed phase react relatively fast within a few minutes at the start of the experiment relative to the slower gas-condensed phase reactions.

Continuous vaporization of the Pb species into the gas phase takes place in the open system through the reactions II.1.c and II.1.g, resulting in PbO (sl)→Pb (g), II.2.c PbS (mat)→PbS (g), and II.1.f PbO (sl)→PbO (g), so the corresponding vectors to the left and down on the Pb$_{matte}$ vs. Pb$_{slag}$ diagram and away from the PbO apex on the PbO-"FeO"-SiO$_2$-phase diagram will always be present during equilibration.

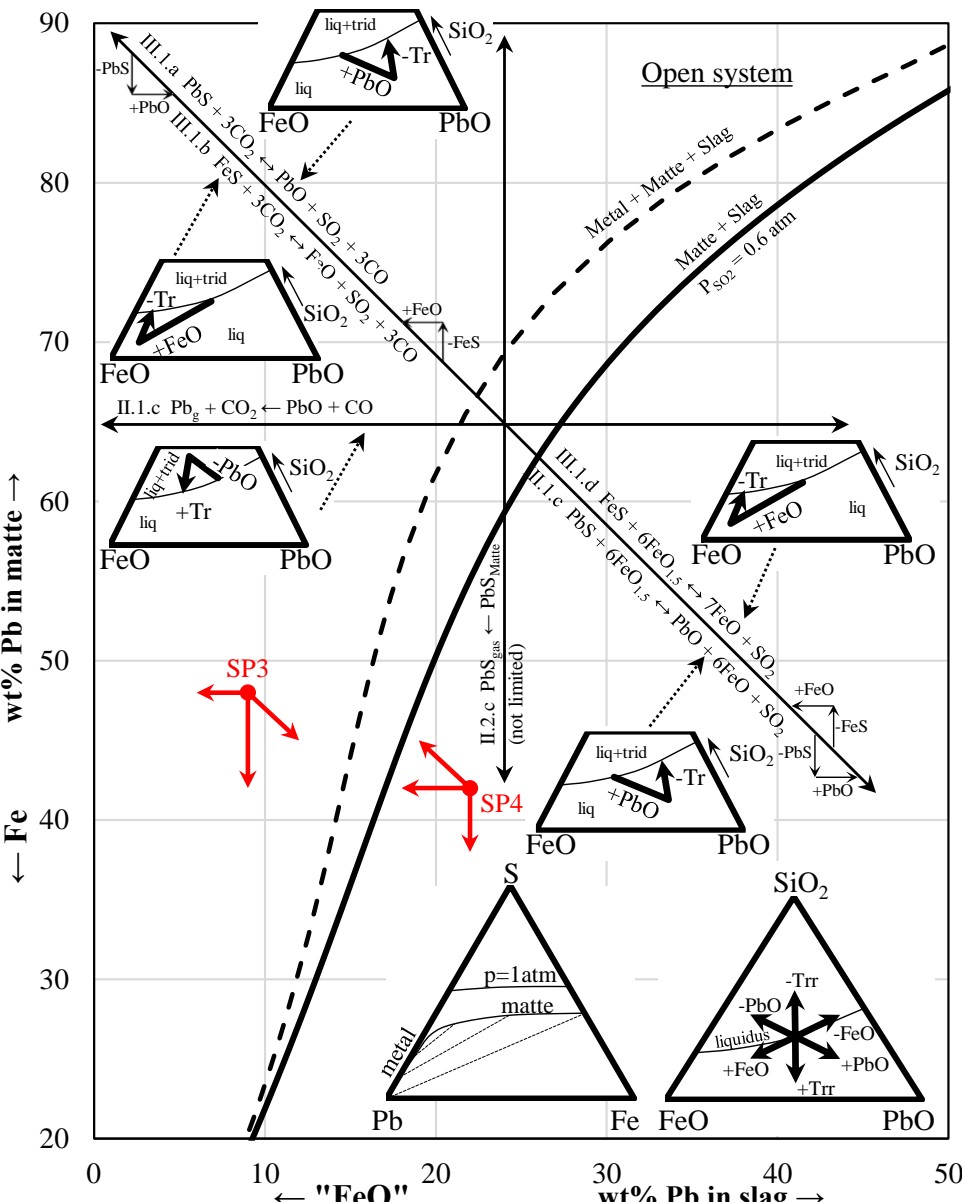

**Figure 8.** Compositional change direction diagrams for $Pb_{matte}$ vs. $Pb_{slag}$ and PbO-"FeO"-$SiO_2$ with the vectors of the compositional changes (red arrows) indicating possible changes in matte and slag compositions corresponding to each reaction in the open system.

Figure 9 presents a schematic illustration of the classification of macro- and micro-locations containing important typical combinations of phase sizes and locations derived from the anticipated interphase reactions and analysis of the mass transfer processes taking place in the open system experiments. The classification of the relative sizes of the phases for the open system is the same as for the closed system, the only difference being that only the large gas phase is relevant to the present study. The II.1 gas–slag can have l-l(t) and l-l(n) combinations; the II.2 gas–matte can only have the l-l combination; the II.4. slag–matte can have l-l, s-l, and l-s, each having "n" and "t" variations. The three-phase gas–slag–matte reactions indicated by "III.1" (see Table 2) can have four size-based combinations—l-l-l, l-s-l, l-l-s, and l-s-s with each of the slag-containing phase combinations having "t" and "n" variations indicating the presence or absence of tridymite in the vicinity of the phase location.

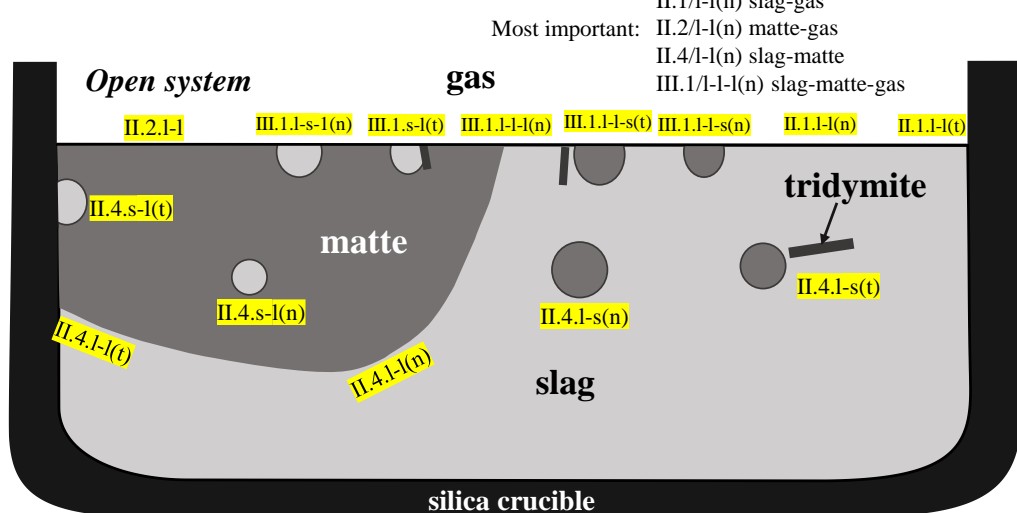

**Figure 9.** Schematic diagram for classification of macro- and micro-locations in the open system.

The entrainment of small slag particles inside or on the large matte particle is possible but rare due to the corresponding physical properties, such as density, surface tension, etc. In contrast, the entrainment of small matte particles in slag is commonly observed.

Since matte does not have the ability to transfer Si or $SiO_2$, then the III.1.l-s-l(t) gas–small slag (with tridymite)—large matte location can achieve local equilibrium dominated by the initial composition of the matte phase. The small slag particle composition will change the following combination of the reactions: III.1.a, b, c, and d; the expected fast mass transfer in the large matte particle will keep the matte phase composition relatively unchanged. The slag and matte compositions measured close to this: III.1.l-s-l(t) location are expected in the preliminary experiments to be close to the true equilibrium condition. Any of the other two-phase locations would be further from the equilibrium; for example, both II.4.l-l(t) and II.4.l-l(n) slag–matte locations would have uncertain $p_{O2}$ and $p_{S2}$ as they are "separated" by the condensed phases from the reactions with the gas phase. The II.1.l-l(t) and II.1.l-l(n) gas–slag locations might not be in equilibrium with the matte phase. The III.1.l-l-l(t) and III.1.l-l-s(t) gas–slag–matte locations are expected to be close to the equilibrium condition, whereas the III.1.l-l-s(t) gas–slag–matte location may be expected to be closer to the initial slag composition in the preliminary short non-equilibrium experiment.

Depending on the slag viscosity, the dissolution of tridymite from the substrate and subsequent $SiO_2$ mass transfer through the slag phase may or may not be sufficiently fast enough to establish the slag–tridymite equilibrium across the whole slag sample. The morphology of the slag–tridymite interface (smooth or having tridymite crystals growing into the slag volume) and the compositional trend of the $SiO_2$ concentration from the slag–substrate interface toward the slag–matte–metal (for a closed system) and gas–slag–matte (for open system experiments) reaction location can indicate if this reaction is incomplete. Crystallization of tridymite can occur on the interface between slag and gas or matte and can block further reaction of slag with the corresponding phase(s).

The definite non-equilibrium locations, such as III.1.l-s-l(n), III.1.l-l-s(n), II.1.l-l(n) and (t), II.4.l-s(n) and (t), and II.4.l-l(n) and (t), in the preliminary experiments can be used to confirm the initial phase compositions and the reactions taking place.

In summary, it is essential to identify the most important reactions that determine mass transfer within and between phases during experiments for the range of conditions investigated. The above analysis of reactions and predicted variations in proportions of phases, phase morphologies, and relative associations established by the starting mixture composition and components, phase physical properties (such as viscosity, density, surface tension, wettability, etc.), and other factors have to be taken into account. For example, when the proportion of the metal phase is lower than that of the matte phase in closed

system experiments, most of the metal phase droplets would be encapsulated within the matte phase, resulting in an interaction between the metal phase and other condensed phases via the matte phase.

The systematic analysis of the trends of phase compositions on the micro- and macro-scales in the samples from the short preliminary experiments with initial compositions of the condensed phases being far from the final equilibrium conditions enabled (i) the key reactions to be identified and confirmed, (ii) the modifications to the experimental technique (in terms of such parameters as starting materials, proportions of phases, the direction of the achievement of equilibrium, temperature regime during experiments, and the duration of equilibration) to then be introduced, and (iii) the methodology, therefore, to finally be developed, as outlined in the following sections.

Once the methodology is established, and the location of the equilibrium curve is approximately established from preliminary experiments, further experiments are planned by selecting the initial composition close to the final equilibrium condition and using all methodology improvements to ensure the equilibrium is achieved.

## 4. Development of Experimental Methodology for the Slag/Matte/Metal/Tridymite (Closed System)

Preliminary equilibration experiments at short equilibration times were performed at 1200 °C (the short treatment times were deliberately selected to obtain non-equilibrium samples to investigate and confirm the reaction taking place during experiments). Typical non-equilibrium microstructures are given in Figure 10, with different locations labeled according to Table 2 and the outlined classification. Most of the locations in these short preliminary non-equilibrium experiments, such as slag–matte–metal without tridymite in the vicinity III.4(n), slag–tridymite II.7, slag–matte without tridymite II.4(n), slag–matte with tridymite II.4(t), slag–metal with tridymite II.5(t), and matte–metal II.6 would achieve only a local equilibrium between two corresponding phases, but only the location slag–matte–metal with tridymite III.4.t was expected to be close to the equilibrium between these four condensed phases.

Note that the formation of tridymite on the interface between slag and matte (Figure 10c) is consistent with the reactions II.4.a $FeS + PbO \rightarrow PbS + FeO$, III.4.c $PbS + 3FeO \rightarrow FeS + 2FeO_{1.5} + Pb$ or II.1.f $PbO (sl) \rightarrow PbO (g)$ associated with precipitation of tridymite. The layer of tridymite would block matte from further reactions with the slag and metal so that only slag–metal reaction II.5 (a), $PbO (sl) + 2FeO (sl) \leftrightarrow 2FeO_{1.5} (sl) + Pb (met)$ would take place in that location. These results were then used to modify the experimental methodology to eliminate this blockage in further final experiments.

The average measured compositions of the phases from equilibration for different times from 15 min to 3 h are given in Table 3.

The results of the experiments using master slag and master matte after various equilibration times are plotted in Figure 11. FactSage predictions for the Pb-Fe-O-S-Si system at the slag–matte–metal–tridymite and slag–matte–metal–tridymite ($p(SO_2)$ = 0.6 atm) equilibria undertaken with the current internal database [4,5] are also provided for comparison. Only the compositions of the slag and matte phases are provided and discussed; the variation in the composition of the metal phase is less significant. The following trends can be observed:

- Pb in both matte and slag decreases as equilibration time progresses, as shown in Figure 11; that may occur due to the vaporization of Pb inside the ampoule followed by re-condensation of Pb droplets in the upper colder part of the ampoule.
- Figure 11a shows that the experimental points give the same trend as FactSage prediction, i.e., Pb in matte increases with increasing Pb in slag.
- Figure 11b,c shows that an increase of Pb in slag leads to a decrease of Fe and S in matte.

- Figure 11j shows the comparison of wt.% $SiO_2$ in slag from experiments with the FactSage prediction. Both show increasing $SiO_2$ in slag with increasing Pb in slag (>15%).
- The sulfur concentration in slag is found to decrease with increasing Pb in slag, as can be observed in Figure 11k.

**Table 3.** Measured phase compositions for the closed slag–matte-Pb metal–tridymite Pb-Fe-O-S-Si system at 1200 °C with master matte and master slag as starting materials.

| No | Initial Mixture | Equilib. Time | Phase | Metal Composition (wt.%) | | | | Phase | Oxide Composition (wt.%) | | | | Fe/SiO$_2$ in Slag | Pb in Slag |
|---|---|---|---|---|---|---|---|---|---|---|---|---|---|---|
| | | | | Pb | Fe | S | Si | | PbO | FeO | S | SiO$_2$ | | |
| 1 | F | 0.25 h | Matte | 64.8 | 17.4 | 17.6 | 0.08 | Slag | 23.9 | 45.7 | 3.8 | 26.5 | 1.34 | 22.2 |
| | | | Metal | 98.3 | 0.13 | 1.53 | 0.02 | Tridymite | | | | >99 * | | |
| 2 | F | 1 h | Matte | 63.8 | 18.2 | 17.7 | 0.10 | Slag | 22.9 | 46.7 | 4.0 | 26.2 | 1.38 | 21.3 |
| | | | Metal | 98.3 | 0.04 | 1.58 | 0.03 | Tridymite | | | | >99 * | | |
| 3 | F | 3 h | Matte | 60.0 | 21.0 | 18.7 | 0.12 | Slag | 20.2 | 49.5 | 4.4 | 25.8 | 1.49 | 18.7 |
| | | | Metal | 98.3 | 0.05 | 1.42 | 0.17 | Tridymite | | | | >99 * | | |
| 4 | E | 0.5 h | Matte | 38.2 | 39.8 | 21.5 | 0.37 | Slag | 15.6 | 54.9 | 5.7 | 23.8 | 1.79 | 14.5 |
| | | | Metal | n/a | n/a | n/a | n/a | Tridymite | | | | >99 * | | |

* Tridymite compositions were not determined accurately due to small size and significant impact of secondary X-ray fluorescence for Fe. Metal compositions have larger uncertainty than matte and slag due to difficulties in quenching. n/a = not analysed.

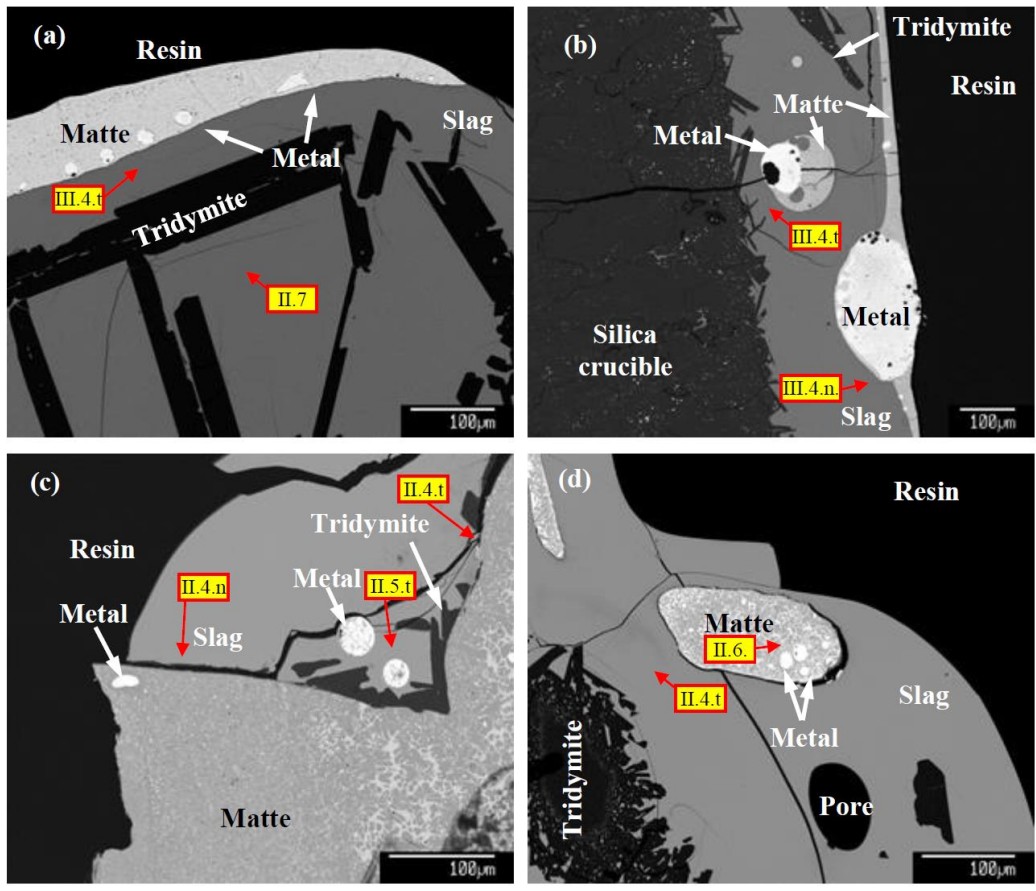

**Figure 10.** Typical microstructures of preliminary short non-equilibrium experiments investigating the closed slag–matte–Pb metal–tridymite Pb-Fe-O-S-Si system.

These changes are consistent with the expected reactions. The key reaction that results in a fast approach to the equilibrium is II.4.a, PbS (matte) + FeO (slag) = PbO (slag) + FeS (matte). The rest of the reactions are limited due to the small proportion of $FeO_{1.5}$ in slag compared to FeO and the limited S concentration in Pb metal. After completion of these sub-reactions within several minutes, the changes in the samples are limited to the growth in the size of the phase droplets.

The clear trend shown by the data indicates that at all times investigated (at each Pb in slag), the condensed phases were close to or in local equilibrium with each other.

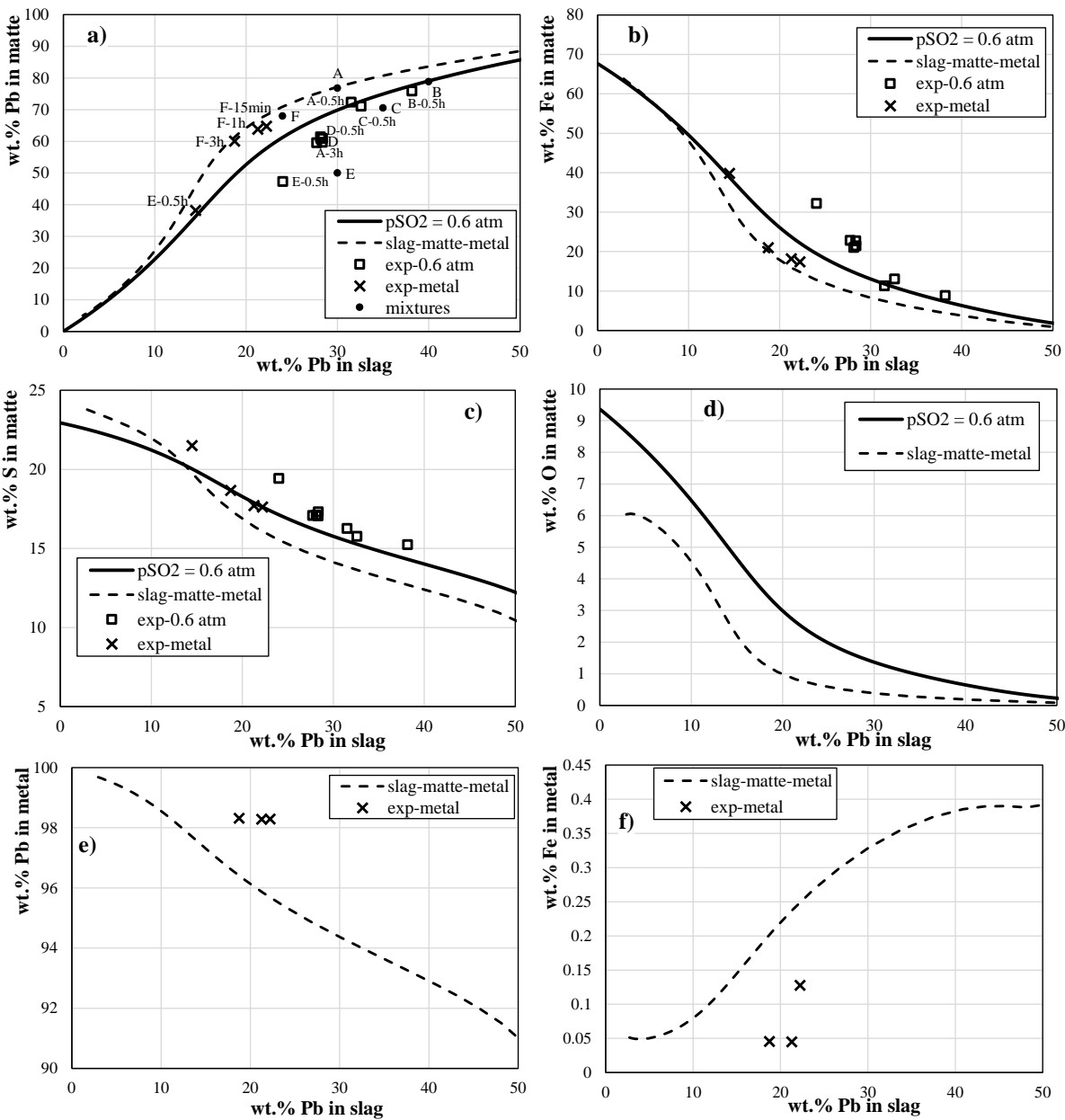

**Figure 11.** *Cont.*

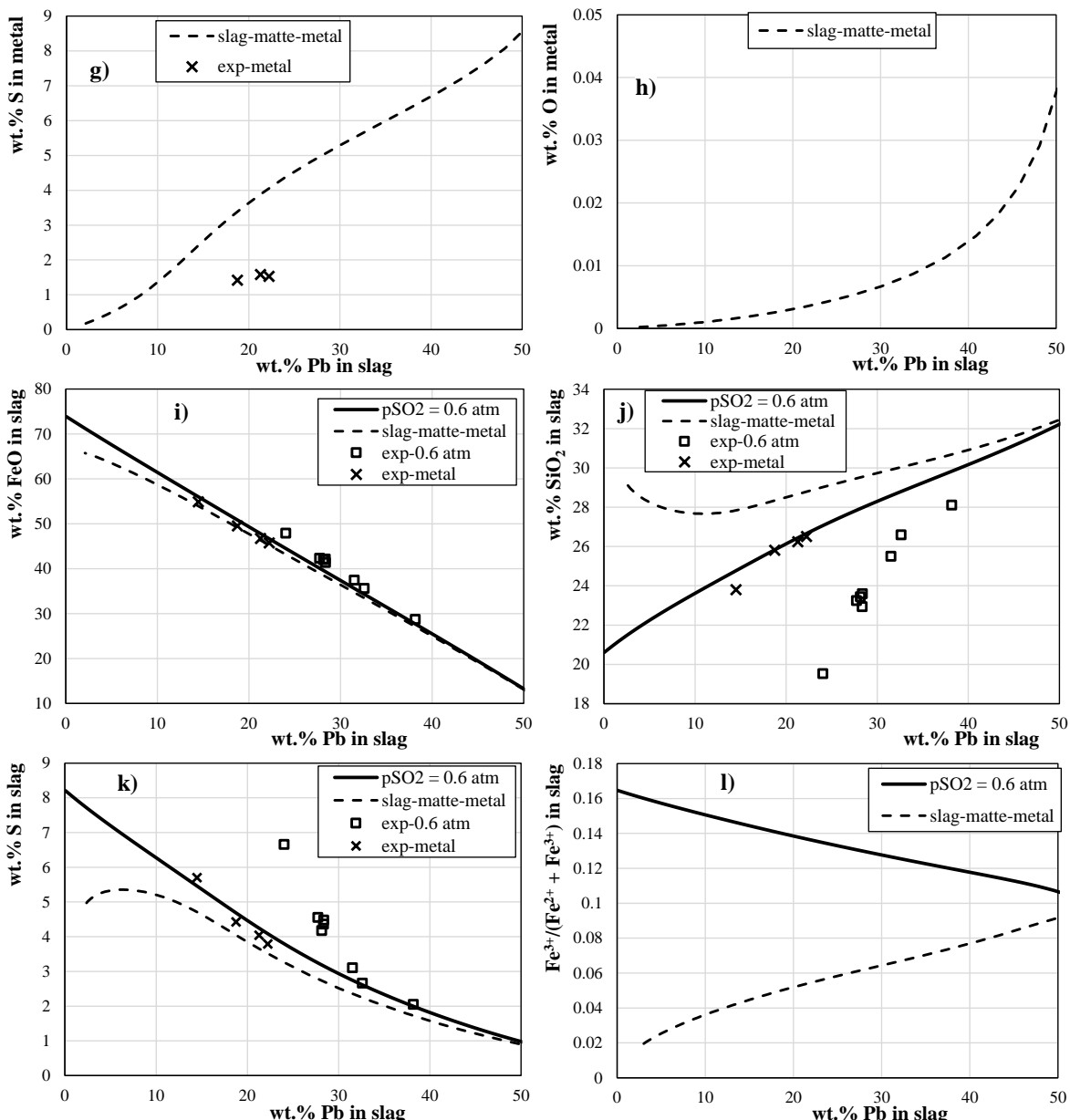

**Figure 11.** Set of graphs describing equilibria in the Pb-Fe-O-S-Si system at 1200 °C and 1 atm of total pressure between slag, matte, tridymite, and metal (dashed lines and squares), or between gas, slag, matte, and tridymite $p(SO_2)$ = 0.6 atm (solid lines and crosses) at various equilibration times between 0.25 and 3 h, including (**a**) wt.% Pb in matte; (**b**) wt.% Fe in matte; (**c**) wt.% S in matte; (**d**) wt.% O in matte; (**e**) wt.% Pb in metal; (**f**) wt.% Fe in metal; (**g**) wt.% S in metal; (**h**) wt.% O in metal; (**i**) wt.% FeO in slag; (**j**) wt.% $SiO_2$ in slag; (**k**) wt.% S in slag; (**l**) fraction of $Fe^{3+}$/total Fe in slag as functions of wt.% Pb in slag. The lines are calculated with the internal FactSage thermodynamic database [4,5]. The label "exp-metal" indicates the experiments in a closed slag–matte–metal system.

## 5. Development of the Experimental Methodology for the Gas–Slag–Matte–Tridymite Semi-Open Equilibrium

### 5.1. Developing the Semi-Open Equilibration Techniques and Testing the Achievement of Equilibrium with the Well Investigated Cu-Fe-O-Si-S System

The experimental approach for the semi-open slag–matte–metal–tridymite equilibrium has been developed based on the assumption that the mass transfer rates of CO, $CO_2$, and $SO_2$ through the small orifice from the outside gas into the ampoule are relatively fast, and the rates of vaporization of PbS, Pb, and PbO from the ampoule through the small orifice

are relatively slow. Several crucible arrangements were tested for the slag–matte–metal–tridymite equilibrium in the Pb-Fe-O-S-Si system at 1200 °C and $p(SO_2)$ = 0.6 atm. The open silica ampoule with silica wool in the middle zone and open top end was found to be suitable for the experiment—the silica wool significantly reduced the loss through the volatilization of the Pb gas species and, at the same time, did not restrict the transfer and reaction of the $CO/CO_2/SO_2$ gas with the slag–matte–tridymite sample.

The ability to achieve the equilibrium between gas and the condensed phase in using a semi-open ampoule with the silica wool in the middle was tested by undertaking a series of experiments involving the equilibration of gas, slag, and matte phases in the Cu-Fe-O-Si-S system at 1200 °C, $p(O_2) = 10^{-8.3}$ atm and $p(SO_2)$ = 0.6 atm that was well characterized previously using the open equilibration technique [34–36]. This test was essential to ensure that the presence of the silica wool in the middle of the ampoule was not blocking the achievement of equilibria between the major gas species and the slag–matte sample located at the bottom of the ampoule. The experiments were conducted by approaching the final equilibrium point from higher, exact, and lower matte grades (% Cu in matte) in the initial starting mixture. The equilibration times varied between 0.5 and 3 h. The results are summarized in Table 4 and plotted in Figure 12.

**Table 4.** Phase compositions for semi-open slag–matte–metal–tridymite experiments in the Cu-Fe-O-S-Si system at 1200 °C, fixed $p(SO_2)$ = 0.6 atm, log $pO_2$ = −8.3 atm.

| Sample | Time, h | Direction | Phase | Matte-Slag Composition, wt.% | | | | |
|--------|---------|-----------|-------|---------|--------|------|---------|-----------|
| | | | | $Cu/Cu_2O$ | Fe/FeO | S | $Si/SiO_2$ | Old Total |
| Cu2H | 0.5 | From high-matte grade | Matte | 73.9 | 4.8 | 21.2 | 0 | 103.0 |
| | | | Slag | 0.9 | 66.1 | 0.6 | 32.4 | 100.6 |
| Cu2H | 3 | From high-matte grade | Matte | 70.7 | 7.3 | 22 | 0 | 102.6 |
| | | | Slag | 0.9 | 66.3 | 1 | 31.8 | 100.6 |
| Cu2E | 0.5 | From exact-equilibrium-matte grade | Matte | 66.6 | 10.3 | 23.1 | 0 | 102.4 |
| | | | Slag | 0.8 | 66.5 | 1.1 | 31.5 | 101.3 |
| Cu2E | 3 | From exact-equilibrium-matte grade | Matte | 64.2 | 12.6 | 23.2 | 0 | 100.9 |
| | | | Slag | 0.8 | 66.6 | 1.4 | 31.1 | 100.8 |
| Cu2L | 0.5 | From low-matte grade | Matte | 57.6 | 18 | 24.3 | 0 | 100.7 |
| | | | Slag | 0.9 | 66.9 | 2 | 30.2 | 101.3 |
| Cu2L | 3 | From low-matte grade | Matte | 60.1 | 15.7 | 24.2 | 0 | 101.6 |
| | | | Slag | 0.9 | 66.7 | 1.8 | 30.5 | 101.2 |

Figure 12 shows that the relationship between the $Fe/SiO_2$ ratio in slag at $p(SO_2)$ = 0.6 atm is reproduced even at a very short equilibration time of 0.5 h. In addition, the relationships between condensed phase compositions (not shown in the figure), i.e., S in matte, S in slag, and Cu in slag as a function of matte grade, follow the equilibrium trend line for $p(SO_2)$ = 0.6 atm. The trend of the matte grade (the Cu/(Cu + Fe + S) weight ratio) is consistent with the system moving toward equilibrium, but 3 h was insufficient to achieve equilibrium since the experiments with different initial mixtures did not merge to one matte grade within this time.

This series of experiments was well investigated before; Cu-Fe-O-Si-S system [34–36] indicates that using the present semi-open approach, it is possible to achieve the equilibrium between the condensed phases and the gas phase corresponding to $p(SO_2)$ = 0.6 atm using the open-top end ampoule and the silica wool in the middle. The rapid interaction between condensed phases and the $CO/CO_2/SO_2$ gas is consistent with the fact that the $SO_2$ concentration in the gas atmosphere was high and, thus, provided a sufficient driving force for the $SO_2$ diffusion through the semi-permeable silica wool barrier. The equilibrium

between the condensed phases and the $p(O_2)$, on the other hand, could not be achieved with the present technique, possibly due to the low $p(O_2)$ value.

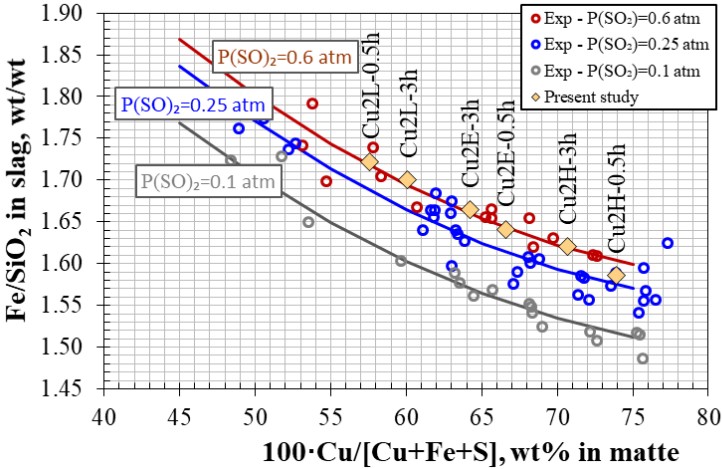

**Figure 12.** $Fe/SiO_2$ in slag as function matte grade in the slag–matte–metal–tridymite experiments in the Cu-Fe-O-S-Si system at 1200 °C, fixed $p(SO_2)$ = 0.6 atm. Present experiments (0.5 and 3 h at $p(SO_2)$ = 0.6 atm) are indicated by yellow diamond symbols. Other data are from experiments using open-type $SiO_2$ substrate for 24 h obtained in previous studies [34–36].

The silica ampoule with the silica wool barrier in the middle zone and the open-top end was, therefore, used in the present study for further semi-open experiments on the gas–slag–matte Pb-Fe-O-S-Si system based on this test series with the previously well investigated Cu-Fe-O-Si-S system [34–36].

### 5.2. Compositional Movements at Short Reaction Times

Short-time preliminary experiments were performed in the Pb-Fe-O-Si-S semi-open system to investigate the changes in the matte and slag compositions during the equilibration. The mass ratio of the sulfide:oxide (matte:slag) in each initial sample was 2:1. The experimental results are summarized in Table 5 and Figure 13; the initial starting mixtures are also indicated in the figure.

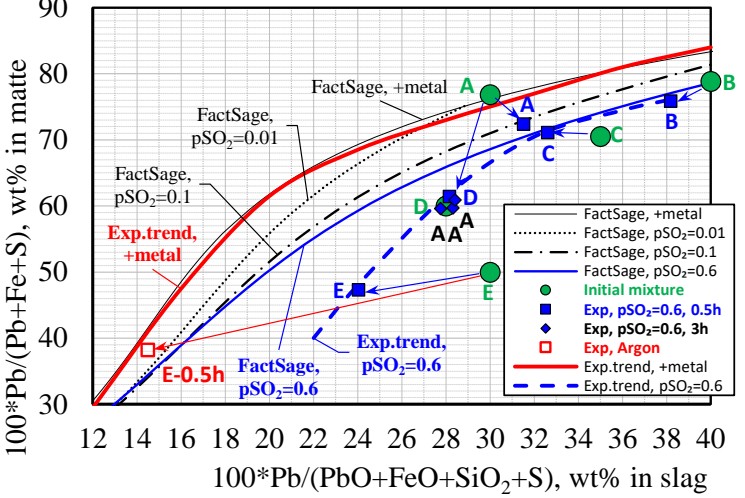

**Figure 13.** Compositional movement within 0.5–3 h for the slag–matte–metal–tridymite equilibrium in the Pb-Fe-O-Si-S system. Green circle = initial starting mixture; Blue square/diamond = after 0.5/3 h equilibration in $CO/CO_2/SO_2/Ar$ atmosphere ($pSO_2$ = 0.6), respectively; Red open square = after 0.5 h equilibration in Ar atmosphere.

**Table 5.** Compositions of the phases obtained in the semi-open slag–matte–metal–tridymite equilibrium experiments in the Pb-Fe-O-Si-S system.

| Mixture | Log $pO_2$ | $pSO_2$ | Time, h | Phase | Composition, wt.% | | | | |
|---------|-----------|---------|---------|-------|---------|-------|-----|---------|-----------|
| | | | | | Pb/PbO | Fe/FeO | S | Si/SiO$_2$ | Old Total |
| A | −8.3 | 0.6 | 0.5 | Matte | 72.4 | 11.3 | 16.3 | 0.0 | 100.0 |
| | | | | Slag | 34.0 | 37.4 | 3.1 | 25.5 | 105.3 |
| B | −8.3 | 0.6 | 0.5 | Matte | 75.9 | 8.8 | 15.2 | 0.0 | 98.6 |
| | | | | Slag | 41.1 | 28.7 | 2.0 | 28.1 | 104.4 |
| C | −8.3 | 0.6 | 0.5 | Matte | 71.1 | 13.0 | 15.8 | 0.1 | 99.2 |
| | | | | Slag | 35.1 | 35.6 | 2.7 | 26.6 | 104.6 |
| D | −8.3 | 0.6 | 0.5 | Matte | 61.4 | 21.0 | 17.1 | 0.4 | 97.9 |
| | | | | Slag | 30.3 | 42.0 | 4.2 | 23.4 | 105.5 |
| E | −8.3 | 0.6 | 0.5 | Matte | 47.3 | 32.2 | 19.4 | 1.1 | 93.8 |
| | | | | Slag | 25.9 | 47.9 | 6.7 | 19.5 | 105.7 |
| A | −8.3 | 0.6 | 3 | Matte | 59.7 | 22.7 | 17.1 | 0.46 | 97.7 |
| | | | | Slag | 30.5 | 42.2 | 4.4 | 22.9 | 104.1 |
| A | −8.3 | 0.6 | 3 | Matte | 60.9 | 21.4 | 17.3 | 0.39 | 97.7 |
| | | | | Slag | 30.6 | 41.4 | 4.5 | 23.6 | 104.8 |
| A | −8.3 | 0.6 | 3 | Matte | 59.6 | 22.8 | 17.1 | 0.44 | 97.7 |
| | | | | Slag | 29.9 | 42.3 | 4.6 | 23.2 | 105.4 |
| E | Argon | n/a | 0.5 | Matte | 38.2 | 39.8 | 21.5 | 0.4 | 96.9 |
| | | | | Slag | 15.6 | 54.9 | 5.7 | 23.8 | 105.1 |

In these experiments, the slag compositions were found to change significantly, while the changes in matte compositions were relatively small. This appears to be due to the high proportion of sulfide (or matte) in the initial mixture; hence, the matte composition is close to that of the initial mixture, while slag composition changes following equilibrium with the gas phase (matte composition governs the equilibration).

For experiments in the $CO/CO_2/SO_2$ atmosphere with $p(SO_2) = 0.6$ atm for 0.5 or 3 h equilibration time, the slag and matte compositions obtained are on the slag–matte–metal–tridymite equilibrium trend line at 1200 °C and $p(SO_2) = 0.6$ atm. For the experiment in an argon atmosphere for 0.5 h equilibration time, the experimental slag and matte compositions lie on the trend line for the slag–matte–metal–tridymite equilibrium at 1200 °C. These results show that equilibration between matte and slag phases with the $p(SO_2)$ can be achieved within a short reaction time (0.5 h).

Most trends in Figure 13 are toward decreasing PbO in slag, which is consistent with a significant effect of the Pb species evaporation. The trend toward higher PbO in slag and lower PbS in matte for one of the experiments (A, 0.5 h) can be explained by the relatively rapid reaction PbS (matte) + FeO (slag) = PbO (slag) + FeS (matte) and the limited time for Pb evaporation. The changes observed in these short experiments confirmed the expected compositional trends.

## 6. Concluding Statements

Following careful analysis of the reaction mechanisms and pathways, experimental methodologies for the systematic accurate measurement of phase equilibria in the Pb-Fe-O-S-Si system have been developed. The achievement of equilibrium in the slag–matte-lead metal–tridymite closed system, and the gas–slag–matte–tridymite open system in $CO/CO_2/SO_2/Ar$ atmospheres has been demonstrated. These are the first reported

systematic accurate phase equilibrium experiments in this system critical for the pyromet-allurgical lead extraction and recycling processes.

The present paper not only outlines the development of the technique for the particular system but also demonstrates a generic approach that includes a few critical stages:

1. Analysis of reactions considering compositions of phases in terms of elements, stable chemical species, and other phase properties; this analysis includes the following steps/components:

    i. Constructing the interphase mass transfer schematic diagram summarising the mass transfer of key elements and key reactions;

    ii. Summarising in one table all selected elementary reactions and reaction steps potentially taking place in the system;

    iii. Constructing the compositional change direction diagram showing the direction of compositional changes in key phases for each reaction;

    iv. Constructing the macro/micro-location diagram identifying the typical locations where the reactions take place within the samples;

2. Short experiments followed by EPMA measurements of compositional profiles to identify macro- and micro-inhomogeneities and to identify and confirm the actual reactions taking place in the system;

3. Longer experiments also followed by the EPMA measurements of macro- and micro-inhomogeneities trends;

4. Modification of the experimental methodology to ensure the achievement of equilibria; and

5. Continuous confirmation of achievement of equilibrium during the experimental program.

Based on the technique developed in the present study, the following industrially important systems are planned for study:

- An extended study of Pb-Fe-Si-O-S at closed and open conditions;
- Effect of copper (Cu-Pb-Fe-Si-O-S)
- Effect of slagging elements (Al, Ca, Mg, Zn)
- Distribution of other minor elements (As, Sn, Sb, Bi, Ag, Au, Ni) between slag, metal, and matte.

**Author Contributions:** Conceptualization, E.J. and P.C.H.; methodology, T.H.; validation and investigation, A.F.-M.; resources, E.J. and P.C.H.; writing—original draft preparation, T.H.; writing—review and editing, M.S.; supervision, E.J. and P.C.H.; project administration, E.J.; funding acquisition, E.J. and P.C.H. All authors have read and agreed to the published version of the manuscript.

**Funding:** The authors would like to thank Aurubis AG (Germany), Boliden (Sweden), Kazzinc Ltd., Glencore (Kazakhstan), Nyrstar (Australia), Outotec Pty Ltd. (Australia), Penoles (Mexica), Umicore NV (Belgium), and Australian Research Council Linkage project LP150100783 for their financial support for this research.

**Data Availability Statement:** The data presented in this study are available on request from the corresponding author. Some of the data may be not publicly available due to conditions of funding.

**Acknowledgments:** The authors are grateful to Suping Huang for assistance with conducting experiments and to the staff of the University of Queensland Centre for Microanalysis and Microscopy (CMM) for their support in maintenance and operation of scanning and electron microprobe facilities in the Centre.

**Conflicts of Interest:** The authors declare no conflict of interest.

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
