# Peer review of "Development of Experimental Techniques for the Phase Equilibrium Study in the Pb-Fe-O-S-Si System Involving Gas, Slag, Matte, Lead Metal and Tridymite Phases"

_processes, doi:10.3390/pr11020372_

Round 1

Reviewer 1 Report

1. Paper should be as the template. (Check abstract)

2. On what basis two critical conditions have been included.

3. Add future prospects in the concluding statements.

4. Fig.2 is missing.

5. Needs more explanation on figure 3.

6. Improve quality image of figure 4.

7. Needs more explanation on figure 12.

8. What is the significance of EPMA measurement.

9. What is Compositions of the phases.

10. The objectives of the work should be clearly stated.

Author Response

  1. Paper should be as the template. (Check abstract)

- The paper has been reformatted according to the template.

  1. On what basis two critical conditions have been included.

- Coexistence with metal corresponds to the minimum p(SO2) for which matte still can exist, while coexistence with p(SO2) = 0.6 atm corresponds to the maximum p(SO2) that can be practically achieved in the laboratory at 1 atm total pressure (the rest 0.4 atm are needed for fixing pO2 with CO-CO2 mixture). All possible industrial conditions therefore will be in between of these two extremes. This paper proves the concept that the system can be reliably studied at both extremes and therefore cover the whole industrially important area.

  1. Add future prospects in the concluding statements.

- The following text has been added:

Based on the technique developed in the present study, the following industrially important systems are planned for study:

- Extended study of Pb-Fe-Si-O-S at closed and open conditions;

- Effect of copper (Cu-Pb-Fe-Si-O-S)

- Effect of slagging elements (Al, Ca, Mg, Zn)

- Distribution of other minor elements (As, Sn, Sb, Bi, Ag, Au, Ni) between slag, metal and matte.

  1. Fig.2 is missing.

- It is unclear why it was missing in the version you received for review. The original version did contain Fig.2. Maybe there was a glitch in the website doc-to-pdf converting.

  1. Needs more explanation on figure 3.

- The whole Table 2 and a page after it are actually explanation for figure 3. We do not believe it should be extended even more. The main idea for figure 3 is providing a visual scheme for reactions listed after it.

  1. Improve quality image of figure 4.

- There could be some blurriness due to image format conversion. The figure has been replaced, and if the problem persists, please note that the supplementary ppt file contains images of high resolution.

  1. Needs more explanation on figure 12.

- Figure 12 illustrates that achievement of equilibrium with p(SO2) (that correlates with Fe/SiO2 in slag) is a much faster process compared to achievement of equilibrium with p(O2) (CO-CO2 mixture). The points in the Pb-free, Cu-containing system reached the equilibrium values for p(SO2)=0.6 in less than 0.5h, while even 3 h were not enough to achieve equilibrium with p(O2), for which all starting compositions would converge. A likely reasons for that are: 1) less mass transfer to the condensed phase is required for processes involving SO2 than for processes involving CO/CO2, 2) the concentration of SO2 in gas is higher than those of CO and CO2. The conclusion for Pb-containing system is that it is possible to study with short-time experiments at fixed high p(SO2), but the p(O2) should be mainly controlled by the initial bulk mixture composition.

  1. What is the significance of EPMA measurement?

- Compared with SEM-EDS, EPMA with WDS detectors can achieve higher accuracy of composition measurements. Compared to other investigation techniques, e.g. laser ablation, XRD, wet chemistry, EPMA has better spatial resolution and allows not only to study each phase composition separately, but even to detect the concentration gradients in each phase that may have them, and to trace possible incomplete reactions within the sample.

  1. What are compositions of the phases?

- Some phase compositions have been listed in Tables 3 and 5. As this paper is devoted to the development of concept, there were not many samples studied. Extensive lists of samples with individual phase compositions will be published in future papers.

  1. The objectives of the work should be clearly stated.

- The aim of the present paper is detailed outline of the development of reliable experimental methodology for the accurate and systematic characterisation of the gas-slag-matte-metal-solid equilibrium in the Pb-Fe-O-S-Si system at controlled conditions. The approach to the development of experimental methodology outlined in the present paper is generic and applicable to the studies of other similar complex systems.

Reviewer 2 Report

The authors have carried out quite interesting work, which is relevant, but at the same time there are the following comments on the text of the article:

1) The thermodynamic program FactSage is used in the work, but there is no information on the developer organization and the country of this program.

2) The following expression is often found in the text: Error! Reference source not found. It is necessary to remove it.

3) Figures 1-13 should be unified according to the size of the drawings themselves, according to the size of fonts, according to axis signatures, designations. It is necessary to bring everything in accordance with the requirements of the journal. To improve their quality.

4) Figure 2 is missing from the article.

5) There are a lot of empty spaces in the article.

6) Tables 1-4 need to be reworked and arranged in accordance with the requirements.

7) The authors do not specify the brand of the electron microscope, the company, the city and the country of the manufacturer.

8) The list of sources is not designed in accordance with the requirements, there are no DOI articles.

Author Response

The authors have carried out quite interesting work, which is relevant, but at the same time there are the following comments on the text of the article:

1) The thermodynamic program FactSage is used in the work, but there is no information on the developer organization and the country of this program.

- Ref. 5-6 contain detailed information for FactSage developers, based in Montreal, Canada.

2) The following expression is often found in the text: Error! Reference source not found. It is necessary to remove it.

- We have checked both the doc file and converted to pdf version for “error”, none was found. Hope that was a temporary glitch of file converting software that will not repeat.

3) Figures 1-13 should be unified according to the size of the drawings themselves, according to the size of fonts, according to axis signatures, designations. It is necessary to bring everything in accordance with the requirements of the journal. To improve their quality.

- As the document has been reformatted to the Journal format, the figures now have the same width. There could be some blurriness due to image format conversion. Some figure have been replaced, and if the problem persists, please note that the supplementary ppt file contains images of high resolution.

4) Figure 2 is missing from the article.

- It is unclear why it was missing in the version you received for review. The original version did contain Fig.2. Maybe there was a glitch in the website doc-to-pdf converting.

5) There are a lot of empty spaces in the article.

- We hope that the reformatted version has less empty spaces.

6) Tables 1-4 need to be reworked and arranged in accordance with the requirements.

- The tables have been reformatted in the new version. Please let us know if any problem persists, and what kind.

7) The authors do not specify the brand of the electron microscope, the company, the city and the country of the manufacturer.

- EPMA JEOL JXA 8200L (trademark of Japan Electron Optics Ltd., Tokyo, Japan)

8) The list of sources is not designed in accordance with the requirements, there are no DOI articles.

- The references have been reformatted and DOIs added where available.

Round 2

Reviewer 1 Report

The paper may be accepted and recommended for publications.

Author Response

The paper may be accepted and recommended for publications.

- All changes in the new revision are according to requests of another reviewer.

Reviewer 2 Report

The attached file contains comments below.

Author Response

1) In Figures 2, 6, 9, it is necessary to reduce the font of the signature at the top, in proportion to the font of the article.

- Thank you for noticing, corrected.

2) In Figures 5, 8, 11, 13, the signature of the OX and OY axes is represented by different font sizes. Also, the signature of the OX axis is between the axis and the axis values, whereas it should be after the axis values, similar to the OY axis.

- Thank you for noticing, corrected (also for fig.1).

3) It is necessary to issue literary references in accordance with the requirements of the journal. According to the style given below: MDPI and ACS Style
- The new reference list has been reformatted in Endnote with MDPI style.
4) In this article, in a literary review, it is necessary to consider the authors' research experience, which is given in the work below:
Kolesnikov A. S., Naraev V.N., Natorhin M.I., Saipov A.A., Kolesnikova O.G. Review of the processing of minerals and technogenic sulfide raw material with the extraction of metals and recovering elemental sulfur by electrochemical methods. Rasayan Journal of Chemistry. 2020. Vol. 13, № 4. P. 2420-2428. http://dx.doi.org/10.31788/RJC.2020.1346102
- The reference has been added.

Round 3

Reviewer 2 Report

It can be seen that the authors have eliminated all the comments and the manuscript has acquired a better quality in accordance with the requirements. I believe that this manuscript can be accepted in this form.